# Effect of preterm birth on early neonatal, late neonatal, and postneonatal mortality in India

**Ajit Kumar Kannaujiya**[1], **Kaushalendra Kumar**[2]*, **Ashish Kumar Upadhyay**[3], **Lotus McDougal**[4], **Anita Raj**[4], **K. S. James**[1], **Abhishek Singh**[2]

**1** International Institute for Population Sciences, Mumbai, India, **2** Department of Public Health & Mortality Studies, International Institute for Population Sciences, Mumbai, India, **3** GENDER Project, International Institute for Population Sciences, Mumbai, India, **4** Center on Gender Equity and Health, University of California San Diego, San Diego, California, United States of America

* kaushal@iipsindia.ac.in

**Data Availability Statement:** Study is based on publicly available secondary Indian DHS data (NFHS), which can be downloaded from https://dhsprogram.com/.

## Abstract

Despite India having a high burden of infant deaths and preterm birth, there is a clear lack of studies documenting association between preterm birth and infant mortality in India. Additionally, existing studies have failed to account for unobserved heterogeneity while linking preterm birth with infant mortality. Hence, the present study examines association of preterm birth with early neonatal death (ENND), late neonatal death (LNND), and postneonatal death (PNND) in India. We used the reproductive calendar canvassed in the cross-sectional National Family Health Survey 2015–16 (NFHS-4) to identify preterm births. We used multivariable logistic regression to examine the associations for all births, most-, second most-, and third most- recent births occurred in five years preceding NFHS-4. We use mother fixed-effect logistic regression to confirm the associations among all recent births. Among all births, preterm births were 4.2, 3.8, and 1.7 times as likely as full-term births to die during early neonatal, late neonatal, and postneonatal periods respectively. Among most recent births, preterm births were 4.4, 4.0, and 2.0 times as likely as full-term births to die during early neonatal, late neonatal, and postneonatal periods respectively. Preterm births were also associated with risk of only ENND, LNND, and PNND among the second most recent births. Preterm births were associated with risk of only ENND and LNND among the third most recent births. Preterm births were also associated with ENND, LNND, and PNND in the mother fixed-effects regressions. This study establishes associations of preterm birth with ENND, LNND, and PNND in India using over 0.2 million births that occurred in 5 years preceding one of the largest population-based representative household surveys conducted in any part of the world. Our findings call for programmatic and policy interventions to address the considerable burden of preterm birth in the country.

## Introduction

Recent estimates suggest that about 3.9 million infant deaths (defined as death before a child's first birthday) occurred globally in 2019 [1]. Of these, approximately 2.4 million are neonatal

**Funding:** This study was funded by the Bill and Melinda Gates Foundation (OPP1179208, PI: AR). The funders had no role in study design, data collection and analysis, decision to publish, or preparation of the manuscript.

**Competing interests:** The authors have declared that no competing interests exist.

deaths (defined as deaths within 28 days of births). The overwhelming majority (80%) of these deaths occur in Sub-Saharan Africa and South Asia. India alone accounts for 20% of neonatal and 17% of infant deaths [1]. India has a high infant mortality rate (IMR), and at 28 infant deaths per 1000 live births, this IMR is higher than other LMICs in the region, including Nepal (26 per 1000 live births), Bangladesh (26 per live births), and Sri Lanka (6 per 1000 live births) [1]. There are also stark variations within India, by state and urban-rural residence. On one hand, Kerala has an IMR of 6 infant deaths per 1,000 live births that resembles that of developed countries like United States of America (6 infant deaths per 1000 live births), United Kingdom, and France (4 infant deaths per 1000 live births) [1, 2]. On the other hand, Uttar Pradesh with an IMR of 64 infant deaths per 1,000 live births is one of the worst rates in the world, and comparable to IMRs seen in Nigeria (74 infant deaths per 1000 live births), Lesotho (68 infant deaths per 1000 live births), and Mali (60 infant deaths per 1000 live births) [1, 3]. There are tremendous discrepancies in IMRs between urban and rural India, as well (29 and 46 respectively), marking the unequal access to and use of care in the country [3].

According to the World Health Organization (WHO), an estimated 15 million births are preterm every year, and this number is rising [4]. India is a major contributor to the global preterm birth numbers; estimates from 2010 and 2014 suggest that India contributed 23–24% of the global average, with nearly one in six (13%-17%) of live births being preterm [5–7]. The prevalence of preterm birth is unevenly distributed within India, ranging from 9% to 14% to 16% for rural Gujarat, rural Tamil Nadu, and rural West Bengal, respectively [8–10]. These estimates not only suggest considerably high burden of preterm births in the country, but also considerable variation in the burden of preterm birth across the different states of India. Such a high burden of preterm births in India assumes significance given global evidence suggests that preterm births are a leading cause of under-five mortality [4, 11].

Despite high rates and stark subnational variations of IMR and preterm births in India, the relationship between preterm births and infant mortality is not fully understood, particularly in terms of age at death in infancy. A limited number of studies in low- and middle-income country (LMIC) contexts have examined these associations. Studies from LMICs such as Malawi, Mozambique, Tanzania, China, and Nepal have found remarkably high early childhood deaths among preterm births [12–17]. An older review of studies from LMICs reported that preterm births are at 2.7 times higher risk for neonatal mortality compared with full term births [18]. A few Indian studies have also examined association between preterm birth and death during infancy. A study specific to a single poorer state in eastern India, Bihar found a significant association between preterm birth and neonatal death [19]. Another study based on clinical outcomes of neonates born within 7 days of public ambulance transport to hospitals across five states of India (Andhra Pradesh, Assam, Gujarat, Karnataka, and Meghalaya) reported association between preterm birth and neonatal death [20]. A cohort study among poor and tribal women from Gujarat, India also reported higher neonatal death among preterm births compared with term births [21]. A couple of other studies show that preterm birth complications are the leading cause of death among the noenates in India [22, 23]. Importantly, a majority of these studies, including the studies from India, are based on small sample, are hospital-based, or are not representative of the broad population. Moreover, none of these studies accounted for unobserved heterogeneity when linking preterm birth with infant death.

Given the high burden of both infant deaths and preterm births in India, and a clear lack of scientific evidence on the association between these two factors, the present study examines the causal association of preterm birth with early neonatal death (ENND), late neonatal death (LNND), and postneonatal death (PNND) using the reproductive calendar canvassed as a part of a nationally representative survey on births in India, the NFHS-4. We separately examine the association of preterm birth with these three because a number of studies have shown that

72% of all infant deaths in India occur within the first 28 days of birth (i.e., the neonatal period), and 76–78% of these neonatal deaths occurred within the first week of life [24–26]. Such a separate analysis is also warranted because the factors associated with deaths during infancy vary by the age of the infant. While birth-related and maternal factors play an important role during the first month of life [27, 28]. environmental and social factors are believed to play a more major role during months 1–11 of life [27, 29]. Additionally, the proportion of deaths due to preterm birth in India is higher in the early than in the late neonatal period [23]. This study can offer important insight into whether preterm birth maintains risk for infant death across the early neonatal, late neonatal, and postneonatal periods, to support guidance for health care follow-up.

## Materials and methods

### Data

This study uses data from NFHS-4, a nationally representative household survey covering over 99% of India's population. NFHS-4 is a cross-sectional survey conducted across all the states and union territories of India. Interviews were completed with 699,686 women age 15–49, with a response rate of 97% [3].

**Ethics and data availability statement.**  Ethical approval for the questionnaire design and data collection were approved by the Institutional Review Boards of both IIPS and ICF [3]. NFHS-4 data are deidentified prior to sharing, and are made publicly available at https://dhsprogram.com/methodology/survey/survey-display-355.cfm. Ethical exemption for this analysis of publicly available, deidentified data was provided by the University of California San Diego Institutional Review Board (180070XX).

**Analytical sample.**  We used reproductive calendar data collected in NFHS-4 to estimate preterm births. The reproductive calendar includes a monthly history of key events such as births, pregnancies, pregnancy terminations, contraceptive use, and reasons for discontinuation of contraception use for a period going back up to 80 months from the interview month. We considered only reproductive histories of five years (up to 59 months) to minimize self-reporting errors (Fig 1).

Our analysis included only those women who had reported at least one live birth in the 5 years before the survey. Of the total 699,686 women interviewed in NFHS-4, 488,112 women reported no pregnancy in the reference period, and hence were excluded from the analysis. The remaining women (211,574) gave births to 257,181 children in five years preceding the survey. Non-singleton births and births with missing information on covariates were also excluded, resulting in an analytic sample size of 240,847 births. We further excluded 703 births occurred in past 7 days as they had not completed the full exposure period for ENND. Likewise, we excluded 3,224 and 49,349 births occurred in past 28 days and in past 12 months as they had not completed the full exposure period for LNND and PNND respectively. So, the final analytical samples for ENND, LNND, and PNND are 240,144, 237,623, and 191,498 births respectively. Details of sample selection are given in Fig 2.

**Dependent variables.**  The dependent variables of the interest are ENND, LNND, and PNND. ENND was coded as '1' if the child died within the first seven days of live birth, and '0' otherwise. Similarly, LNND was coded as '1' if the child died between the 8th to twenty-eighth day of live birth, and '0' otherwise. PNND was coded as '1' if the child died between 29th day and 1st birthday, and '0' otherwise.

**Key independent variable.**  The key independent variable of interest is preterm birth. Reproductive calendar data collected in NFHS-4 were used to identify preterm births. Preterm births are those that occurred before 9 months of pregnancy [30]. Preterm birth was coded as

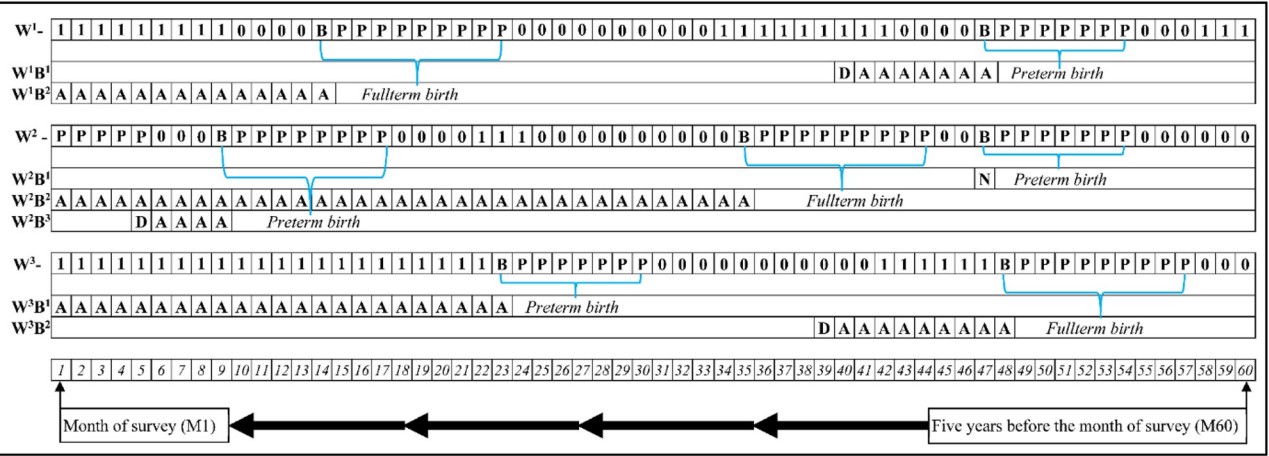

**Fig 1. Reproductive calendar showing estimation of preterm birth and mortality outcomes.** Note. W-Women, P- Pregnant, B- Birth, 0- not using any contraception, 1- using contraception, A- Alive, N- early-/ late- neonatal death, D- postneonatal death, One box represents one month.

'1' if the birth occurred before 9 months of pregnancy, and '0' otherwise. Fig 1 demonstrates the estimation of preterm birth.

**Control variables.** A number of maternal, child, and household variables are associated with early childhood mortality [17, 31–35]. In our analysis, mother-related variables included mother's age at conception (<20 years, 20–24 years, 25–29 years, ≥30 years), mother's height (<145 cm, ≥145 cm), and mother's schooling (no schooling, primary, secondary, or higher). Index child-related variables included birth order and interval (First birth order, birth orders 2 or 3 and birth interval <24 months, birth orders 2 or 3 and birth interval ≥24 months, birth orders ≥4 and birth interval <24 months, birth orders ≥4 and birth interval ≥24 months), delivery through c-section (no, yes), wanted birth (no, yes), and sex of the child (male, female). Household-level variables included wealth quintiles (poorest, poorer, middle, richer, richest), religion (Hindu, Muslim, others), caste (scheduled caste, scheduled tribe, other backward class, others), state-region (north, central, east, northeast, west, south), and urban-rural residence. The details of state-region variable are given in S1 Table.

## Methods

The reproductive calendar structure included in NFHS-4 allows for unique analysis of causal relationships, as the progression of exposures and outcomes is sequenced over 60 months for each child and his/her mother. Fig 1 demonstrates the reproductive calendar for three women–$W^1$, $W^2$, and $W^3$. $W^1$ had two births in last five years—one preterm ($B^1$) and the other fullterm ($B^2$). $B^1$ survived for 7 months and died in the eighth month. So, by definition, $B^1$ died in the postneonatal period. On the other hand, $B^2$ survived the first year of life. $W^2$ had three births in the past five years. While $B^1$ and $B^3$ were preterm births, $B^2$ was fullterm. $B^1$ died during the neonatal period, $B^2$ survived the first year of life, and $B^3$ died during the postneonatal period. The reproductive calendar also allows for examining relationships by sequence of births, such as most-, second most-, third most- recent births. For example, $B^3$, $B^2$, and $B^1$ are the most-, second most-, and third most- recent birth for $W^2$. As the outcome variable is binary, we used multivariable logistic regression to examine the association of preterm birth with ENND, LNND, and PNND in India. First, we examined the association of preterm birth with the three mortality outcomes using all births in the five-years preceding NFHS-4.

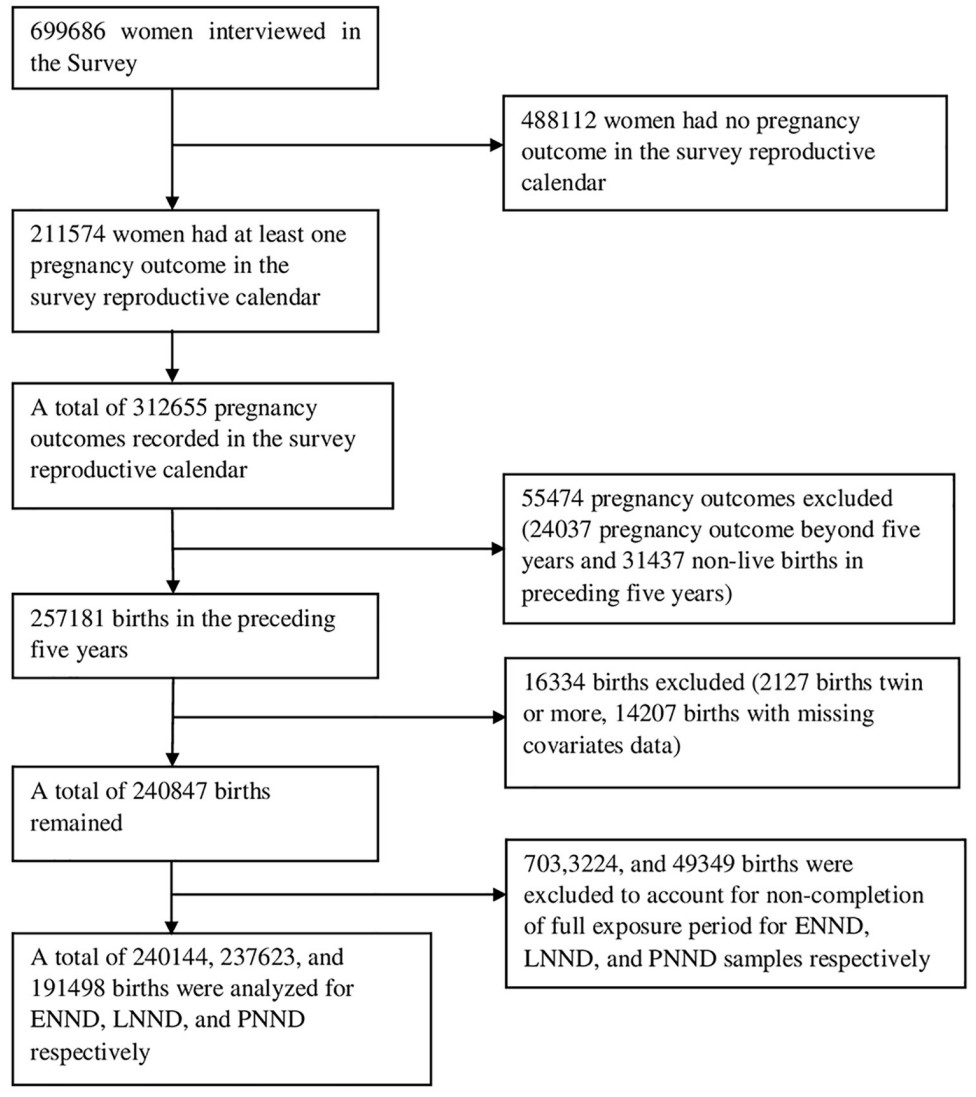

**Fig 2. Sample selection process.**

We then examined the association of preterm birth with the three mortality outcomes separately for the most-, the second most-, and the third most- recent births.

We also estimated mother fixed-effects multivariable logistic regression models to account for the unobserved heterogeneity, which may lead to omitted-variable bias [32]. All mother-, household- and residence- related observed, and unobserved heterogeneity will cancel out and only heterogeneity related to index birth and mother's age will remain in the fixed-effects regressions.

Finally, we did three sensitivity analyses to examine the robustness of our results. Existing research has shown that gestational age of preterm birth may be associated with mortality outcomes [12, 36]. To address this issue, we disaggregated preterm births into two–early preterm (gestational age ≤7 months) and late preterm birth (gestational age = 8 months). We then re-estimated the regression models with the mortality outcomes as independent variables and index birth preterm (no, early preterm, late preterm) as the dependent variable, adjusting for other control variables. Second, research has shown that accessibility to birth and neonatal

services and quality of such services may vary by urban-rural residence [37]. To address this issue, we re-estimated the regression models separately for urban and rural areas. Finally, we re-estimated the regression models for the most recent births only to examine whether the inclusion of maternal and child care program related variables, such as ≥4 antenatal care (ANC) visits (no, yes), ≥2 Tetanus Toxoid (TT) injections (no, yes), and consumed ≥100 Iron Folic Acid (IFA) tablets/equivalent syrup (no, yes), changes the association between preterm birth and the mortality outcomes. We restricted the analysis to most recent births as the afore-mentioned maternal and child care program related variables are available only for the most recent births in NFHS-4.

Since NFHS-4 used a multistage sampling design, appropriate sampling weights were used in estimations. Appropriate adjustments were also made for the complex survey design employed in NFHS-4. All the statistical analyses were conducted in Stata 15.1.

## Results

### Descriptive results

Descriptive statistics of births occurred in five years preceding NFHS-4 are shown in Table 1. Two percent, 0.4%, and 1.0% of the births died in the early neonatal, late neonatal, and post-neonatal periods respectively. Around 7% of births in the early neonatal, late neonatal, and postneonatal sample were preterm. The percent distribution of all the control variables were similar across the three samples. Seventy-two percent of births in the early neonatal and late neonatal subsamples were most recent births. Twenty-four percent and 3% were second and third- most recent births. In contrast, 65% and 30% of births in the postneonatal sample were most recent and second most recent births. Four percent of births were third most recent in this sample. Preterm births ranged between 7% among most recent births and 9% in third most recent births in early neonatal, late neonatal, and postneonatal samples (S2 Table). Two percent, 0.2%, and 1.0% of most recent births died in the early neonatal, late neonatal, and postneonatal periods, respectively. In contrast, 9%, 2%, and 3% of third most recent births died in the early neonatal, late neonatal, and postneonatal periods respectively.

Among all recent births, 8% of preterm births died within 7 days of birth. In comparison, only 2% of the full-term births died within 7 days of birth. Likewise, 1.0% and 0.4% of the pre-term and full-term births died during the late neonatal period respectively. Similarly, 2% and 1% of the preterm and full-term births died during the postneonatal period respectively. Irre-spective of the sequence of birth, a higher percentage of preterm births died in the early neona-tal and late neonatal periods. For example, 24% of preterm third most recent births died within 7 days of birth compared with only 8% of full-term third most recent births. Thirteen percent of preterm second most recent births died within 7 days of birth compared with only 4% of full-term second most recent births. Among the most- and second most- recent births, a higher percentage of preterm births died in the postneonatal period. In contrast, a higher per-centage of full-term births died during the postneonatal period among the third most recent births. The details are shown in S3 Table.

### Multivariable logistic regression results

The adjusted odds ratios showing association between preterm birth and risk of deaths in the early neonatal, late neonatal, and postneonatal periods are shown in Fig 3. Among all recent births, preterm births were 4.2 times as likely as full-term births to die within 7 days of birth. Most recent preterm births, second most recent preterm births, and third most recent preterm births were 4.4 times, 3.7 times, and 3.7 times as likely as their counterparts to die within 7 days of birth respectively. Likewise, among all recent births, preterm births were 3.8 times as

**Table 1. Descriptive statistics of births included in the three samples, NFHS-4, India, 2015–16.**

| | ENND sample | LNND sample | PNND sample |
| --- | --- | --- | --- |
| | **(n = 240 144)** | **(n = 237 623)** | **(n = 191 498)** |
| **Death** | | | |
| No | 234306 (97.6%) | 236616 (99.6%) | 189223 (99.0%) |
| Yes | 5838 (2.4%) | 1007 (0.4%) | 2275 (1.0%) |
| **Index birth preterm** | | | |
| No | 223631 (93.2%) | 221297 (93.2%) | 178614 (93.4%) |
| Yes | 16513 (6.8%) | 16326 (6.8%) | 12884 (6.6%) |
| **Birth order and Birth interval** | | | |
| First birth order | 93020 (40.2%) | 92031 (40.2%) | 74520 (40.5%) |
| Birth orders 2 or 3 and interval <24 months | 29653 (12.8%) | 29391 (12.8%) | 23944 (12.9%) |
| Birth orders 2 or 3 and interval ≥24 months | 80048 (33.4%) | 79162 (33.4%) | 62878 (33%) |
| Birth orders ≥4 and interval <24 months | 9386 (3.5%) | 9311 (3.5%) | 7810 (3.7%) |
| Birth orders ≥4 and interval ≥24 months | 28037 (10.1%) | 27728 (10.1%) | 22 346 (10.1%) |
| **Index child c-section** | | | |
| No | 207167 (82.8%) | 205032 (82.8%) | 166208 (83.3%) |
| Yes | 32977 (17.2%) | 32591 (17.2%) | 25290 (16.7%) |
| **Index birth wanted** | | | |
| No | 19404 (8.1%) | 19168 (8.1%) | 14 839 (7.8%) |
| Yes | 220740 (91.9%) | 218455 (91.9%) | 176659 (92.2%) |
| **Sex of child** | | | |
| Male | 124665 (52.1%) | 123343 (52.1%) | 99169 (51.9%) |
| Female | 115479 (47.9%) | 114280 (47.9%) | 92329 (48.1%) |
| **Mother's age at conception** | | | |
| <20 years | 30465 (14.1%) | 30234 (14.1%) | 25467 (14.7%) |
| 20–24 years | 104704 (45.4%) | 103610 (45.5%) | 83576 (45.4%) |
| 25–29 years | 67295 (27.2%) | 66533 (27.1%) | 52692 (26.6%) |
| ≥30 years | 37680 (13.3%) | 37246 (13.3%) | 29763 (13.3%) |
| **Mother's height** | | | |
| <145cm | 27699 (11.9%) | 27405 (11.9%) | 21954 (11.9%) |
| ≥145cm | 209608 (86.4%) | 207415 (86.4%) | 167325 (86.5%) |
| Refused/Others/Missing | 2837 (1.7%) | 2803 (1.7%) | 2219 (1.6%) |
| **Mother's schooling** | | | |
| No schooling | 73977 (29.6%) | 73297 (29.7%) | 60515 (30.3%) |
| Primary | 34 984 (14.0%) | 34632 (14.0%) | 28303 (14.3%) |
| Secondary or Higher | 131183 (56.4%) | 129694 (56.3%) | 102680 (55.4%) |
| **Caste** | | | |
| Scheduled Caste | 45248 (21.6%) | 44762 (21.6%) | 35862 (21.5%) |
| Scheduled Tribe | 48758 (10.6%) | 48198 (10.5%) | 38893 (10.4%) |
| Other Backward Class | 93739 (44.0%) | 92772 (44.1%) | 74748 (44.1%) |
| Others | 52399 (23.8%) | 51891 (23.8%) | 41995 (24.0%) |
| **Religion** | | | |
| Hindu | 172757 (78.5%) | 170906 (78.5%) | 137316 (78.3%) |
| Muslim | 38069 (16.7%) | 37674 (16.7%) | 30510 (16.9%) |
| Others | 29318 (4.8%) | 29043 (4.8%) | 23672 (4.8%) |
| **Wealth quintiles** | | | |
| Poorest | 63076 (25.1%) | 62417 (25.1%) | 50619 (25.3%) |
| Poorer | 57046 (22.0%) | 56424 (22.0%) | 45362 (21.9%) |

(*Continued*)

**Table 1.** (Continued)

| | ENND sample | LNND sample | PNND sample |
|---|---|---|---|
| | **(n = 240 144)** | **(n = 237 623)** | **(n = 191 498)** |
| Middle | 48000 (19.9%) | 47500 (19.9%) | 38202 (19.8%) |
| Richer | 39908 (18.3%) | 39509 (18.3%) | 31872 (18.3%) |
| Richest | 32114 (14.7%) | 31773 (14.7%) | 25443 (14.7%) |
| **Urban-rural residence** | | | |
| Urban | 56873 (28.2%) | 56316 (28.2%) | 45798 (28.6%) |
| Rural | 183271 (71.8%) | 181307 (71.8%) | 145700 (71.4%) |
| **State-region** | | | |
| North | 44510 (13.0%) | 44055 (13.0%) | 35335 (13.0%) |
| Centre | 68819 (26.7%) | 68002 (26.7%) | 54157 (26.4%) |
| East | 50268 (25.6%) | 49768 (25.6%) | 40257 (25.6%) |
| Northeast | 35580 (3.7%) | 35245 (3.7%) | 28937 (3.8%) |
| West | 17066 (12.9%) | 16872 (12.9%) | 13660 (13.0%) |
| South | 23901 (18.1%) | 23681 (18.1%) | 19152 (18.2%) |
| **Sequence of births** | | | |
| Most recent birth | 173706 (72.3%) | 171197 (72.0%) | 125280 (65.4%) |
| Second most recent birth | 57737 (24.0%) | 57725 (24.3%) | 57518 (30.0%) |
| Third most recent birth | 8154 (3.4%) | 8154 (3.4%) | 8153 (4.3%) |
| Fourth and higher most recent birth | 547 (0.2%) | 547 (0.2%) | 547 (0.3%) |

Note. Early neonatal death (ENND), late neonatal death (LNND), and postneonatal death (PNND).

likely as full-term births to die in the late neonatal period. Most recent preterm births, second most recent preterm births, and third most recent preterm births were 4.0 times, 3.4 times, and 2.8 times as likely as their counterparts to die in the late neonatal period respectively. Preterm births were at much higher risk of dying during postneonatal period compared with full-term births among all recent, most recent, and second most recent births. Full multivariable logistic regression results are shown in the S4–S6 Tables. As an additional sensitivity check, we also assessed our models using neonatal deaths (death in 0–28 days of life) (NND) and infant deaths (death in the first year of life) (IND). Preterm births were also associated with higher chances of deaths in the neonatal and infancy periods (S1 Fig).

## Mother fixed-effects multivariable logistic regression results

The mother fixed-effects regression results for all recent births are shown in Table 2. Preterm births were associated with higher risk of dying in the early neonatal, late neonatal, and post-neonatal periods in the mother fixed-effects regressions. Preterm births were 7.6 times as likely as their counterparts to die within 7 days of birth. Likewise, preterm births were 5.8 times as likely as their counterparts to die in the late neonatal period. Similarly, preterm births were 2.3 times more likely to die in the postneonatal period compared with full-term births. While the odds ratios have become bigger in mother fixed-effects regressions, the sample sizes have reduced considerably. Compared to their counterparts, c-section births were less likely to die in the early neonatal period. Births reported unwanted by mothers were 2.2–2.9 times as likely as those who were reported as wanted to die in the early neonatal, late neonatal, and postneo-natal periods. Preterm births were 7.9 and 6.8 times as likely as full-term births to die during neonatal and infancy periods respectively (S7 Table).

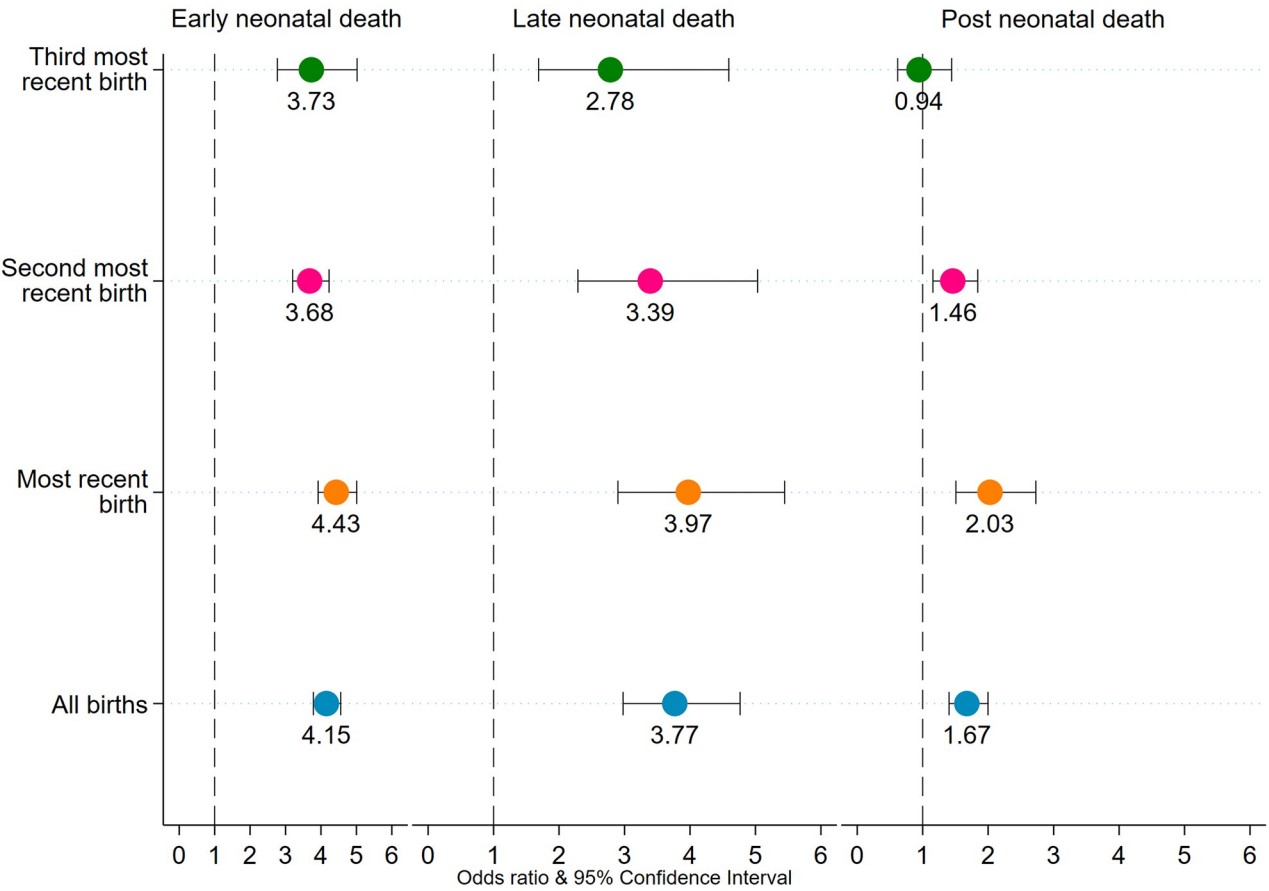

**Fig 3. Adjusted odds ratio of ENND, LNND, and PNND for preterm birth, NFHS-4, India, 2015–16.** Note. 1. Odds ratios are significant at p<0.05. ENND Early Neonatal Death, LNND Late Neonatal Death, PNND Post Neonatal Death. Odds ratio are adjusted for birth order, index child c-section, index birth wanted, sex of child mother's age at conception, mother's height, mother's schooling, caste, religion, wealth quintiles, urban-rural residence, and state-region.

### Sensitivity analyses results

Early preterm births were strongly associated with ENND and LNND among all births and among each of three sequence of births (Fig 4). Late preterm birth was also associated with ENND, LNND, and PNND among all births and among each of three sequence of births. However, the odds of dying among early preterm births were 7–10 times higher compared with the odds of dying among the late preterm births. While early preterm births were statistically associated with PNND among all births and among each of three sequence of births, late preterm births were associated with PNND among all births and the most recent births. The second sensitivity analysis dealt with estimating the association of preterm birth and ENND, LNND, and PNND separately for urban and rural areas (Fig 5). Preterm births were associated with ENND and LNND in both urban and rural areas. While preterm births were associated with PNND among all birth, most recent- and second most- recent births in rural areas, preterm births were associated with PNND only among all birth and most recent births in urban areas. Finally, the robustness analysis on most recent births indeed suggests that the association between preterm births and the five mortality outcomes–ENND, LNND, NND, PNND, and IND–does not change when we include maternal-care programme related factors, such as

**Table 2. Mother fixed-effects multivariable logistic regression results for ENND, LNND, and PNND for all births in last five years, NFHS-4, India, 2015–16.**

| Variable & category | ENND | LNND | PNND |
|---|---|---|---|
| | (n = 7399) | (n = 1465) | (n = 2307) |
| | Odds ratio (95%CI) | Odds ratio (95%CI) | Odds ratio (95%CI) |
| **Index birth preterm** | | | |
| No (reference) | 1.00 | 1.00 | 1.00 |
| Yes | 7.60* (5.93, 9.74) | 5.76* (3.53, 9.39) | 2.31* (1.48, 3.63) |
| **Birth order and birth interval** | | | |
| First birth order | 5.40* (3.96, 7.36) | 5.90* (3.12, 11.18) | 1.70* (1.01, 2.85) |
| Birth orders 2 or 3 and interval <24 months | 1.44* (1.09, 1.91) | 2.66* (1.49, 4.74) | 1.11 (0.69, 1.79) |
| Birth orders 2 or 3 and interval ≥24 months | 2.03* (1.55, 2.66) | 4.07* (2.28, 7.24) | 1.49 (0.94, 2.37) |
| Birth orders ≥4 and interval <24 months | 0.84 (0.66, 1.07) | 1.66 (0.95, 2.88) | 0.88 (0.60, 1.29) |
| Birth orders ≥4 and interval ≥24 months (reference) | 1.00 | 1.00 | 1.00 |
| **Index child c-section** | | | |
| No (reference) | 1.00 | 1.00 | 1.00 |
| Yes | 0.48* (0.37, 0.63) | 0.70 (0.33, 1.48) | 0.64 (0.34, 1.18) |
| **Index birth wanted** | | | |
| No | 2.85* (2.25, 3.62) | 2.43* (1.42, 4.14) | 2.21* (1.47, 3.32) |
| Yes (reference) | 1.00 | 1.00 | 1.00 |
| **Sex of child** | | | |
| Male (reference) | 1.00 | 1.00 | 1.00 |
| Female | 0.62* (0.55, 0.69) | 0.55* (0.43, 0.71) | 1.10 (0.91, 1.33) |
| **Mother's age at conception** | | | |
| <20 years | 1.06 (0.83, 1.35) | 2.29* (1.25, 4.17) | 1.39 (0.90, 2.14) |
| 20–24 years (reference) | 1.00 | 1.00 | 1.00 |
| 25–29 years | 0.83 (0.67, 1.03) | 1.36 (0.90, 2.08) | 0.91 (0.62, 1.33) |
| ≥30 years | 0.63* (0.42, 0.95) | 1.50 (0.62, 3.63) | 0.46 (0.19, 1.07) |

**Notes**:

* p < 0.05; CI: Confidence interval, early neonatal death (ENND), late neonatal death (LNND), and postneonatal death (PNND).

≥4 antenatal care (ANC) visits (no, yes), ≥2 Tetanus Toxoid (TT) injections (no, yes), and consumed ≥100 Iron Folic Acid (IFA) tablets/equivalent syrup (no, yes), into the regression models (Fig 6).

## Discussion

This is the first Indian study that establishes association of preterm birth and ENND, LNND, and PNND using the reproductive calendar canvassed as part of NFHS-4. Children who were born preterm were four times more likely to die in the early neonatal or late neonatal periods, and more than 1.7 times more likely to die in the postneonatal period.

While our findings are in line with research from both high and lower income countries [12–19, 38], we build upon the limitations of previous smaller sample and less nationally representative research by demonstrating robust association between preterm birth and death during infancy using over 0.2 million births that occurred in 5 years preceding one of the largest population-based representative household surveys conducted in any part of the world. Our use of the temporal and child-specific reproductive calendar, the mother fixed-effects along with the multivariable regression analyses differentiated by sequence of births allows us to move statements of association into those of causality, an important

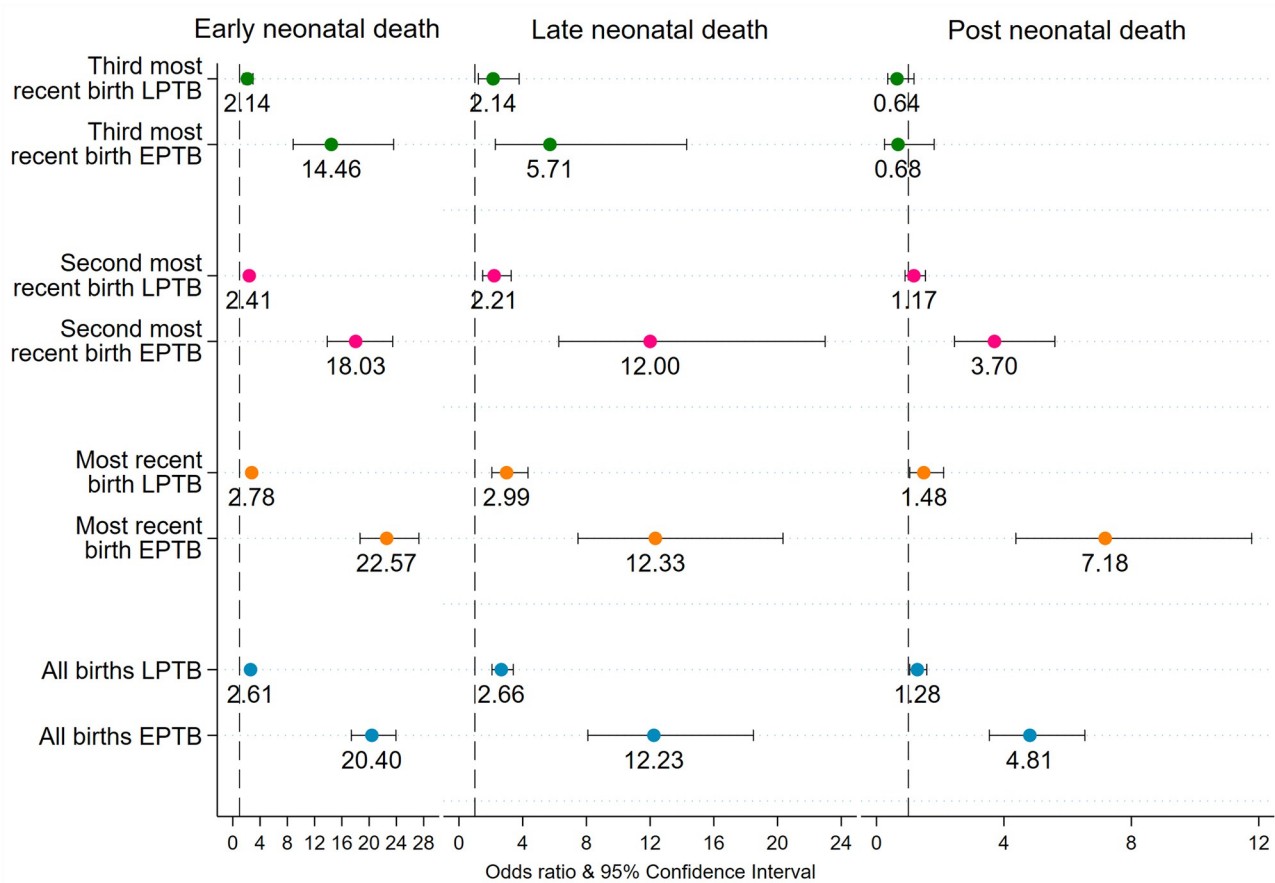

**Fig 4. Adjusted odds ratio of ENND, LNND, and PNND for early preterm birth (EPTB) and late preterm birth (LPTB), NFHS-4, 2015–16.** Note. 1. Odds ratios are significant at p<0.05. ENND Early Neonatal Death, LNND Late Neonatal Death, PNND Post Neonatal Death. Odds ratio are adjusted for birth order, index child c-section, index birth wanted, sex of child mother's age at conception, mother's height, mother's schooling, caste, religion, wealth quintiles, urban-rural residence, and state-region.

improvement over prior studies. Additionally, the study demonstrates that preterm birth maintains risk for infant mortality across the full first year of life, though risk remains greatest in the neonatal period.

When adjusting for the impact of individual mothers who had multiple births during the window of analysis, the risk of deaths during the early neonatal, late neonatal, and postneonatal periods was even larger. Although these results support the findings of our main models, mother fixed-effects models come with some cost. Since the estimation is based on mothers who have given at least two births in the 5 years preceding NFHS-4, the sample size reduces drastically leading to lower precision of the estimates. Moreover, mother fixed-effects models in our study are based on sibling pairs in which one sibling has an outcome different from that of the other sibling. Thus, we ended up with considerably smaller samples with greater representation of poorer, rural, and less educated mothers, though these factors were accounted for in all our regression models. Nonetheless, findings from these analyses document the protectiveness of C-sections and the increased risk of infant mortality in cases of unintended pregnancy. These findings reinforce the importance of access to reproductive and maternal health care as means of addressing infant mortality in India.

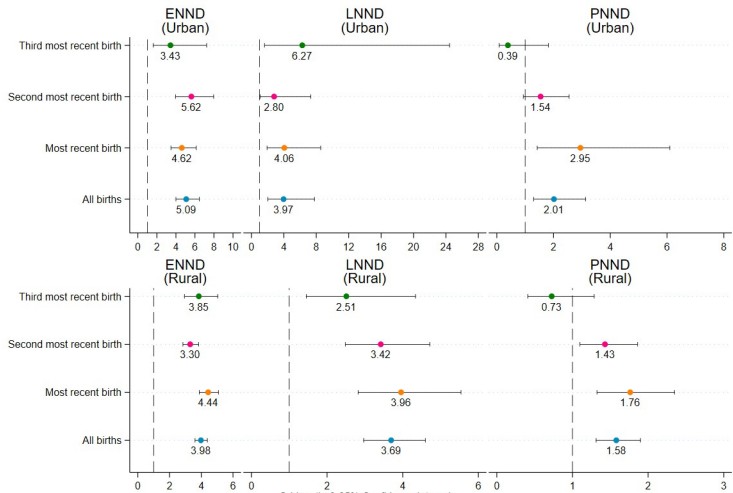

**Fig 5. Adjusted odds ratio of ENND, LNND, and PNND for preterm birth stratified by urban-rural residence, NFHS-4, 2015–16.** Note. 1. Odds ratios are significant at p<0.05. ENND Early Neonatal Death, LNND Late Neonatal Death, PNND Post Neonatal Death. Odds ratio are adjusted for birth order, index child c-section, index birth wanted, sex of child mother's age at conception, mother's height, mother's schooling, caste, religion, wealth quintiles, and state-region.

Sensitivity analysis confirmed a positive association of early and late preterm births with ENND and LNND among all births and each sequence of birth; though the risk of ENND and LNND was much higher among the early preterm births. Early preterm births were also associated with PNND, which is an important contribution of this study. Preterm births were also associated with ENND and LNND in both urban and rural areas. A key finding that also

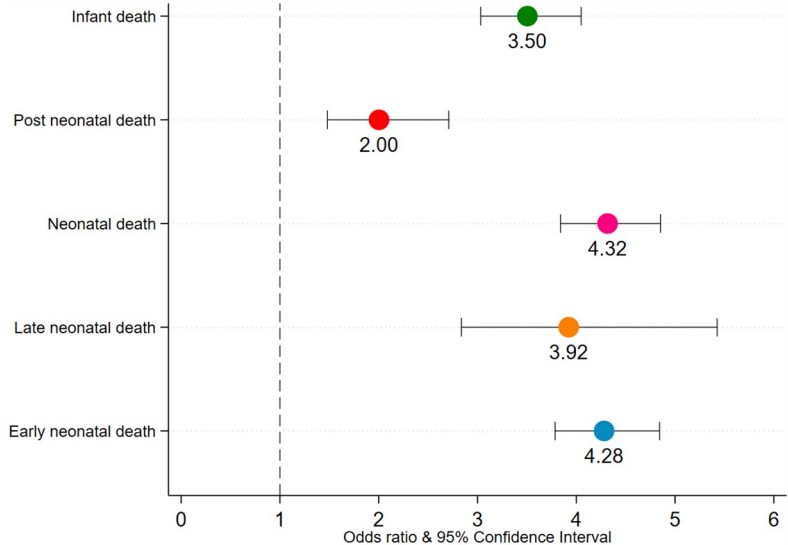

**Fig 6. Adjusted odds ratio of ENND, LNND, NND, PNND, and IND for preterm birth among most recent births, NFHS-4, India, 2015–16.** Note. 1. Odds ratios are significant at p<0.05, ENND Early Neonatal Death, LNND Late Neonatal Death, PNND Post Neonatal Death, NND Neonatal Death, IND Infant Death. Odds ratio are adjusted for birth order, index child C-section, > = 4 ANC visit, >100 IFA tablets, > = 2 TT injection, index birth wanted, sex of child mother's age at conception, mother's height, mother's schooling, caste, religion, wealth quintiles, urban-rural residence, and state-region.

deserves mention is the protective effect of recommended antenatal care, net of preterm birth and other control variables. Findings related to the most recent births indicate that births of mothers who availed 4 or more antenatal visits were less likely than births of mothers who did not avail the recommended visits to die during the early neonatal period. Likewise, births of mothers who received 2 or more TT injections were less likely than births of mothers who did not receive 2 or more TT injections to die during the early neonatal or postneonatal periods. Importantly, consumption of 100 or more IFA tablets had a protective effect on neonatal and infant mortality. While, all the three components– 4 or more antenatal visits, 2 or more TT injections, and consumption of 100 or more IFA tablets–had protective effect on neonatal mortality, only receiving 2 or more TT injections and consumption of 100 or more IFA tablets had protective effect on infant mortality. Our findings are consistent with prior research [39], and indicate that ensuring recommended antenatal care for all women may partially offset the adverse consequences of preterm birth in the early neonatal, late neonatal, and postneonatal periods.

There are important study limitations to note, including the definition of preterm births employed in our study. According to the WHO, any birth before 37 completed weeks of gestation, or fewer than 259 days since the first day of the women's last menstrual period is termed as preterm birth [40]. Unfortunately, the NFHS collects information on duration of gestation in months, not weeks; assessing gestational age in months rather than weeks has a precedent in India [19]. Additionally, NFHS does not assess the causes of preterm births, limiting our ability to make cause-specific recommendations based on our findings. Another key limitation is potential reporting and recall bias on the part of mother. This is particularly relevant for the information collected in the reproductive calendar. Since no research has systematically examined the accuracy of reproductive calendar for estimating preterm births, it is difficult to estimate the effect of such biases on our models. Our analysis of third most recent births and mother fixed-effects are based on small samples which may have affected the precision of the estimates. We could not estimate the associations for 4$^{th}$ and higher most recent births due to sample size limitations. We also could not include several important child-level variables, such as birth weight and cause of death, in the regression models due to data limitations: birth weight was missing for 22% of the births, and NFHS-4 did not collect any data on causes of deaths. While survival success of preterm births may depend on quality of health care services available to them, we could not adjust our model estimates for health care services variables due to data limitations.

To conclude, this research provides compelling evidence on a causal linkage between preterm birth and death during infancy. About 25 million children are born every year in India [41]. As per our estimates, about 1.7 million of these births are preterm, and thus at higher risk of death during early neonatal, late neonatal, and postneonatal periods. There is, therefore, a need to formulate and rigorously implement appropriate policies to prevent and offset the serious consequences of preterm birth if India is to achieve Sustainable Development Goal 3.2 (end preventable deaths of newborns and children under 5 years of age). Prior to delivery, high quality antenatal care, including counselling on healthy diet and optimal nutrition, fetal measurements, screening for multiple fetuses, identifying and managing infections, and ongoing fetal monitoring are key [42]. Better access to family planning methods and increased woman's empowerment may also help in reducing the burden of preterm birth in the country [4]. Following delivery, simple low-cost interventions, such as kangaroo mother care, clean cord care and breastfeeding, can substantially reduce the mortality burden of preterm births [43]. Recognizing the importance of these interventions, the Government of India has taken important steps towards establishing *Special Newborn Care Units* (SNCUs) in district hospitals [44]. In addition, UNICEF has developed a *Facility Based Newborn Care Database* for the

National Health Mission (NHM) to monitor and track small and sick newborn in real time [45]. As the mortality risk associated with preterm births extend beyond the neonatal period [13, 18], a refocus on early childhood along with interventions in the immediate neonatal period is also needed to improve survival of preterm newborns during infancy and later childhood. It is also critical to monitor the quality of care received by these vulnerable babies to ensure respectful and appropriate care and avert morbidity and mortality in health facility settings. Finally, there is a need for more research on the more long-term impacts of preterm birth on child health indicators, such as under-five mortality, physical growth, and cognitive development, in Indian children.

## Supporting information

**S1 Fig. Adjusted odds ratio of NND and IND for preterm birth, NFHS-4, India, 2015–16.**
Note: 1. Odds ratios are significant at p < 0.05. 2. Odds ratios are adjusted for birth order, index child c-section, index birth wanted, sex of child, mother's age at conception, mother's height, mother's schooling, caste, religion, wealth quintiles, urban-rural residence, and state-region. Neonatal deaths (NND), infant deaths (IND).
(DOC)

**S1 Table. States included in various categories of state-region.**
(DOC)

**S2 Table. Percentage of preterm birth and ENND, percentage of preterm birth and LNND, and percentage of preterm birth and PNND by sequence of birth among births in the past five years, NFHS-4, India, 2015–16.** Notes: 1. Fourth most recent births in the past five years were excluded due to very small sample sizes. 2. 95% confidence intervals are shown in the parenthesis. 3. Early neonatal death (ENND), late neonatal death (LNND), and postneonatal death (PNND).
(DOC)

**S3 Table. ENND, LNND, and PNND by preterm birth according to sequence of birth among births in the past five years, NFHS-4, India, 2015–16.** Note: 1. Fourth most recent births in the past five years were excluded due to very small sample sizes. 2. Early neonatal death (ENND), late neonatal death (LNND), and postneonatal death (PNND).
(DOC)

**S4 Table. Adjusted odds ratio of early neonatal deaths (ENND) for preterm birth, NFHS-4, India, 2015–16.** Note: OR: odds ratio; * p < 0.05; CI: Confidence interval.
(DOC)

**S5 Table. Adjusted odds ratio of late neonatal deaths (LNND) for preterm birth, NFHS-4, India, 2015–16.** Note: OR: odds ratio; * p < 0.05; CI: Confidence interval.
(DOC)

**S6 Table. Adjusted odds ratio of post neonatal deaths (PNND) for preterm birth, NFHS-4, India, 2015–16.** Note: OR: odds ratio; * p < 0.05; CI: Confidence interval.
(DOC)

**S7 Table. Mother fixed-effects multivariable binary logistic regression results for neonatal deaths (NND) and infant deaths (IND) for all births in last five years, NFHS-4, India, 2015–16.** Note: OR: odds ratio; * p < 0.05; CI: Confidence interval.
(DOC)

## Author Contributions

**Conceptualization:** Ajit Kumar Kannaujiya, Kaushalendra Kumar, Abhishek Singh.

**Data curation:** Ajit Kumar Kannaujiya, Ashish Kumar Upadhyay.

**Formal analysis:** Ajit Kumar Kannaujiya, Ashish Kumar Upadhyay.

**Funding acquisition:** Anita Raj.

**Investigation:** Ashish Kumar Upadhyay, Lotus McDougal.

**Methodology:** Ajit Kumar Kannaujiya, Ashish Kumar Upadhyay, Abhishek Singh.

**Project administration:** K. S. James.

**Supervision:** Kaushalendra Kumar.

**Writing – original draft:** Ajit Kumar Kannaujiya, Ashish Kumar Upadhyay, Abhishek Singh.

**Writing – review & editing:** Ashish Kumar Upadhyay, Lotus McDougal, Anita Raj, K. S. James, Abhishek Singh.

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
