## [Decision Letter · Decision Letter 0]

9 Dec 2021

PGPH-D-21-00856

Effect of preterm birth on early neonatal, late neonatal, and postneonatal mortality in India

Dear Dr. Kumar,

Thank you for submitting your manuscript to PLOS Global Public Health. After careful consideration, we feel that it has merit but does not fully meet PLOS Global Public Health’s publication criteria as it currently stands. Therefore, we invite you to submit a revised version of the manuscript that addresses the points raised during the review process.

We look forward to receiving your revised manuscript.

Kind regards,

Zohra S. Lassi, PhD

Academic Editor

Journal Requirements:

1. Please amend your detailed Financial Disclosure statement. This is published with the article, therefore should be completed in full sentences and contain the exact wording you wish to be published.

i) Please include all sources of funding (financial or material support) for your study. List the grants (with grant number) or organizations (with url) that supported your study, including funding received from your institution. 

ii). State the initials, alongside each funding source, of each author to receive each grant.

iii). State what role the funders took in the study. If the funders had no role in your study, please state: “The funders had no role in study design, data collection and analysis, decision to publish, or preparation of the manuscript.”

iv). If any authors received a salary from any of your funders, please state which authors and which funders.

2. Please send a completed 'Competing Interests' statement, including any COIs declared by your co-authors. If you have no competing interests to declare, please state "The authors have declared that no competing interests exist". Otherwise please declare all competing interests beginning with the statement "I have read the journal's policy and the authors of this manuscript have the following competing interests:"

3. Please note that your Data Availability Statement is currently missing a direct link to access each database. If your manuscript is accepted for publication, you will be asked to provide these details on a very short timeline. We therefore suggest that you provide this information now, though we will not hold up the peer review process if you are unable.

4. We do not publish any copyright or trademark symbols that usually accompany proprietary names, eg (R), (C), or TM  (e.g. next to drug or reagent names). Therefore please remove all instances of trademark/copyright symbols throughout the text, including No ®, Birth orders ≥4 order and interval ≥24 months®, Yes ®, Male ®, 20-24 years ® on pages 14 and 15.

5. We have noticed that you have uploaded supporting information but you have not included a list of legends.  Please add a full list of legends for all supporting information files (including figures, table and data files) after the references list.

Additional Editor Comments (if provided):

Reviewers' comments:

Reviewer's Responses to Questions

**Comments to the Author**

1. Does this manuscript meet PLOS Global Public Health’s publication criteria? Is the manuscript technically sound, and do the data support the conclusions? The manuscript must describe methodologically and ethically rigorous research with conclusions that are appropriately drawn based on the data presented.

Reviewer #1: Yes

Reviewer #2: Yes

2. Has the statistical analysis been performed appropriately and rigorously?

Reviewer #1: Yes

Reviewer #2: Yes

3. Have the authors made all data underlying the findings in their manuscript fully available (please refer to the Data Availability Statement at the start of the manuscript PDF file)?

Reviewer #1: Yes

Reviewer #2: Yes

4. Is the manuscript presented in an intelligible fashion and written in standard English?

Reviewer #1: Yes

Reviewer #2: Yes

5. Review Comments to the Author

Reviewer #1: Kannaujiya et al., have presented an interesting study on the association of preterm birth with early, late and post neonatal death in India. The study relies on the data from NFHS-4 survey and reports that children born preterm were four times more likely to die in the early or late neonatal periods. The authors have also listed the limitations of the study, and this study addresses some key questions. However, it will be great if the authors can have more clarity for the following points:

1. Preterm birth can be caused due to multiple factors and can have long-term effect. Here the authors need to clarify if the neonates were (a) small for gestational age, (b) IUGR, (c) extreme, moderate or late preterm? The authors have specified that the NFHS-4 has time reported in months and not weeks, but even from that information the extreme or moderate preterm can be estimated.

2. Were the preterm spontaneous, or due to preeclampsia or PPROM? This classification is not addressed in the manuscript.

3. The association with hospitalization and neonatal care is also not accounted for in the manuscript. It will also be good to report the neonatal death in urban versus rural settings and the neonatal care available.

4. Are only singleton pregnancies considered in this study?

5. Why was the maternal age considered from 15-49 years and not the traditionally used threshold of 18 years? The chances of preterm is higher for women <18 years of age.

6. The IRB numbers are not specified in the manuscript. It will be good to include them in the methods section.

Reviewer #2: This is the first study on analysis of association of preterm birth and ENND, LNND, and PNND with the findings are very useful for India, especially for the LICs and LMICs. The huge sample size of over 0.2 million births that occurred in 5 years preceding one of the largest population-based representative household surveys is very convincing as seen in the findings of study. Preterm births were also associated with higher risk of dying in the early neonatal, late neonatal, and post-neonatal periods in the family fixed-effects regressions.

Children who were born preterm were four times more likely to die in the early neonatal or late neonatal periods, and more than 1.7 times more likely to die in the post-neonatal period. There are many significant mother related factors as control variables in data analysis to see how important these variables are in contribution to the ENND, LNND, and PNND. ENND among preterm births as compared with no preterm births.

However, some comments that authors should take into considerations for possible revision or a further writing:

- Study design was not mentioned (cross sectional or case control study design?) and I do suggest that author should clarify it in the Abstract and Materials and Methods section. Defined study design will help reader know how they could understand about the aim of study as well as the methods for statistical analysis of data.

- My question is why the child related factors and their death causes of children (before and after neonatal period) were not included as control variables as well as modified risk factors in the study. Authors should add them in this manuscript (if any) and put these data in your additional analysis, because child related factors are also important to see other child biological factors leading these child deaths together with their mother related factors. Otherwise, author should say this is limitation of study that these data were not available.

- The findings from Table 2 (Mother fixed-effects multivariable logistic regression results for ENND, LNND, and PNND for all births in last five years, NFHS-4, India, 2015-16) are very important as risk factors of preterm birth (Birth order, child C-section, sex of child....) and they should be shortly presented in the abstract, not only showing single association between neonatal mortality and preterm birth.

- The author calculated OR (see Table 2) to see risk factors of neonatal mortality as preterm birth or mother related factors etc. Is this study design therefore defined as case control for data analysis? Please explain to clarify.

- Do authors have any other evidences to show Ethical approval, for this study, by the Institutional review Boards of both IIPS and ICF? Looking at the website: https://dhsprogram.com/ seems not enough information about ethical consideration.

6. PLOS authors have the option to publish the peer review history of their article (what does this mean?). If published, this will include your full peer review and any attached files.

**Do you want your identity to be public for this peer review?** For information about this choice, including consent withdrawal, please see our Privacy Policy.

Reviewer #1: No

Reviewer #2: No

---

## [Decision Letter · Decision Letter 1]

1 Apr 2022

PGPH-D-21-00856R1

Effect of preterm birth on early neonatal, late neonatal, and postneonatal mortality in India

Dear Dr. Kumar,

Thank you for submitting your manuscript to PLOS Global Public Health. After careful consideration, we feel that it has merit but does not fully meet PLOS Global Public Health’s publication criteria as it currently stands. Therefore, we invite you to submit a revised version of the manuscript that addresses the points raised during the review process.

We look forward to receiving your revised manuscript.

Kind regards,

Zohra S. Lassi, PhD

Academic Editor

Journal Requirements:

1. Your co-authors, Ashish Kumar Upadhyay (ashu100789@gmail.com), Anita Raj (anitaraj@ucsd.edu), and K S James (ksjames@iipsindia.ac.in), have not confirmed authorship of the manuscript. We have resent them the authorship confirmation email; however please check that the above email address for them is correct and follow up personally to ensure they confirm. Please note that we cannot pass your manuscript to Production until we have received confirmations from all co-authors.

Just in case your co-authors are having difficulty confirming their authorship, you may advise them to send us an email at globalpubhealth@plos.org and we will confirm their authorship on the authors' behalf.

Additional Editor Comments (if provided):

Reviewers' comments:

Reviewer's Responses to Questions

**Comments to the Author**

1. If the authors have adequately addressed your comments raised in a previous round of review and you feel that this manuscript is now acceptable for publication, you may indicate that here to bypass the “Comments to the Author” section, enter your conflict of interest statement in the “Confidential to Editor” section, and submit your "Accept" recommendation.

Reviewer #1: All comments have been addressed

Reviewer #2: All comments have been addressed

2. Does this manuscript meet PLOS Global Public Health’s publication criteria? Is the manuscript technically sound, and do the data support the conclusions? The manuscript must describe methodologically and ethically rigorous research with conclusions that are appropriately drawn based on the data presented.

Reviewer #1: Yes

Reviewer #2: Yes

3. Has the statistical analysis been performed appropriately and rigorously?

Reviewer #1: Yes

Reviewer #2: N/A

4. Have the authors made all data underlying the findings in their manuscript fully available (please refer to the Data Availability Statement at the start of the manuscript PDF file)?

Reviewer #1: Yes

Reviewer #2: Yes

5. Is the manuscript presented in an intelligible fashion and written in standard English?

Reviewer #1: Yes

Reviewer #2: Yes

6. Review Comments to the Author

Reviewer #1: The authors have done a great work in revising the manuscript and addressing all the comments. I have minor comments related to the work:

1. In Table 2, what does the term 'reference' suggest? It's repeated a number of times in the table, but is not indicated in the legends.

2. In the multivariable logistic regression model, do the authors adjust for mother's age at the time of birth?

Reviewer #2: (No Response)

7. PLOS authors have the option to publish the peer review history of their article (what does this mean?). If published, this will include your full peer review and any attached files.

**Do you want your identity to be public for this peer review?** For information about this choice, including consent withdrawal, please see our Privacy Policy.

Reviewer #1: **Yes: **Priyanka Baloni

Reviewer #2: **Yes: **Thang V. Vo

---

## [Decision Letter · Decision Letter 2]

4 Jun 2022

Effect of preterm birth on early neonatal, late neonatal, and postneonatal mortality in India

PGPH-D-21-00856R2

Dear Mr Kumar,

We are pleased to inform you that your manuscript 'Effect of preterm birth on early neonatal, late neonatal, and postneonatal mortality in India' has been provisionally accepted for publication in PLOS Global Public Health.

Best regards,

Julia Robinson

Staff Editor

Reviewer Comments (if any, and for reference):

Reviewer's Responses to Questions

**Comments to the Author**

1. If the authors have adequately addressed your comments raised in a previous round of review and you feel that this manuscript is now acceptable for publication, you may indicate that here to bypass the “Comments to the Author” section, enter your conflict of interest statement in the “Confidential to Editor” section, and submit your "Accept" recommendation.

Reviewer #1: All comments have been addressed

2. Does this manuscript meet PLOS Global Public Health’s publication criteria? Is the manuscript technically sound, and do the data support the conclusions? The manuscript must describe methodologically and ethically rigorous research with conclusions that are appropriately drawn based on the data presented.

Reviewer #1: Yes

3. Has the statistical analysis been performed appropriately and rigorously?

Reviewer #1: Yes

4. Have the authors made all data underlying the findings in their manuscript fully available (please refer to the Data Availability Statement at the start of the manuscript PDF file)?

Reviewer #1: Yes

5. Is the manuscript presented in an intelligible fashion and written in standard English?

Reviewer #1: Yes

6. Review Comments to the Author

Reviewer #1: The authors have addressed all my comments.

7. PLOS authors have the option to publish the peer review history of their article (what does this mean?). If published, this will include your full peer review and any attached files.

**Do you want your identity to be public for this peer review?** For information about this choice, including consent withdrawal, please see our Privacy Policy.

Reviewer #1: **Yes: **Priyanka Baloni
